# Morphological and Anatomical Analysis of the Internodes of a New Dwarf Variant of Moso Bamboo, *Phyllostachys edulis* f. *exaurita*

**DOI:** 10.3390/plants12091759

**Published:** 2023-04-25

**Authors:** Ruofei Zha, Tianguo Chen, Qingnan Liu, Qiang Wei, Feng Que

**Affiliations:** 1State Key Laboratory of Tree Genetics and Breeding, Co-Innovation Center for Sustainable Forestry in Southern China, Key Laboratory of National Forestry and Grassland Administration on Subtropical Forest Biodiversity Conservation, College of Biology and the Environment, Nanjing Forestry University, Nanjing 210037, China; ruofeizha9702@163.com (R.Z.); liuqingnan24@163.com (Q.L.); weiqiang@njfu.edu.cn (Q.W.); 2Bamboo Research Institute, Nanjing Forestry University, Nanjing 210037, China; 3Changzhou Agricultural Comprehensive Technology Extension Center, Changzhou 213022, China; tianguochen2023@163.com

**Keywords:** Moso bamboo, dwarfing variant, morphology, anatomy, *primary thickening growth*, *Phyllostachys edulis* f. *exaurita*

## Abstract

The lack of mutants due to the long periods between flowering of bamboo plants is one of the limiting factors inhibiting research progress in the culm development of bamboo plants. In this study, a stable new dwarf variant of *Phyllostachys edulis* (Moso bamboo), *Phyllostachys edulis* f. *exaurita T. G. Chen*, was discovered and was characterized morphologically, anatomically, and physiologically. The height, diameter at breast height, number of internodes, length and wall thickness of internodes, length, width and number of parenchyma cells of internodes, and morphology of the wide-type (WT) and dwarf variant vascular bundles were compared. The height of the variant was only 49% that of the WT Moso bamboo. It was concluded that the decrease in internode number and length was the cause of dwarfism in *P. edulis* f.* exaurita*. The decreased internode length was the result of a decrease in cell number and cell length in the internode. In addition, the laws of change of internode length, internode thickness, cell length, and cell number differed between the WT Moso bamboo and the variant. Furthermore, lower IAA and zeatin concentrations were detected in the buds of the variant. These results suggest that *P. edulis* f. *exaurita* is a variant with inhibited primary thickening growth, which is valuable for interpretating the molecular mechanisms underlying the primary thickening growth of bamboo that are still largely unknown.

## 1. Introduction

Bamboos belong to the subfamily Bambusoideae of the grass family Poaceae, and they are classified into woody and herbaceous bamboos. Compared to other members of Poaceae, woody bamboos have a tree-like culm with more lignification [1,2]. Bamboos have an excellent ability to sequester carbon and contribute greatly to climate change mitigation [3]. In addition, the bamboo culms can be used as a building material, and the bamboo shoots are a very healthy vegetable [3]. As an important non-timber forestry plant, bamboo generates a high economic production value each year [4].

Accordingly, more and more attention has been paid to the basic biological research on bamboo plants. With the rapid development of technology, high-throughput sequencing has greatly promoted research on the basic biology of bamboo plants [5,6,7]. In contrast, the loss of germplasm resources (variants) has become a limitation in bamboo plant research. To date, although a number of bamboo variants have been reported, few of them are stable variants [8]. *P. edulis* ‘*Pachyloen*’ is a stable variant of Moso bamboo with abnormal primary thickening growth, in which a thickened culm wall and a small pith cavity are observed [5]. Further comprehensive analysis revealed that the enlarged shoot apical meristem (SAM) with increased cell number of *P. edulis* ‘*Pachyloen*’ may cause the abnormal primary thickening growth, eventually leading to the abnormal culm morphologies [5]. *Phyllostachys nidularia* f. *farcta* is another stable variant of *P. nidularia*. Compared to the WT bamboo plants, *P. nidularia* f. *farcta* is dwarfed and has lower biomass [9]. Systematic characterization of the variant found that variations in morphology and cell number of SAM are the main cause of the phenotype of the variant [9]. Some research advances have also been made by studying other bamboo variants, such as *Pseudosasa japonica* var. *tsutsumiana* (a slow-growing variant of *P. japonica*) and *P. edulis* f. *tubaeformis* (a dwarf variant of Moso bamboo) [10,11]. A systematic comparison between the stable variant and the WT bamboo plant is helpful to promote basic research on bamboo biology.

Moso bamboo (*Phyllostachys edulis*) is an important economic bamboo species with the largest cultivation area in China (~4.43 million hectares) [12]. It is known for its rapid growth, which can reach 114.5 cm/day [13]. The development process of Moso bamboo culm includes three main distinct stages, which are primary thickening growth of the underground shoot, rapid growth of the unearthed shoot, and the secondary cell wall thickening growth of the newly formed culm [10]. In recent years, the rapid growth of Moso bamboo has been extensively studied. Hormones (including gibberellin, cytokinin, auxin, and abscisic acid), mechanical pressure, and environmental temperature were considered to be the main factors contributing to the high growth rate of bamboo internodes [13,14,15]. In contrast, little attention has been paid to the primary thickening growth of the underground shoot. In Moso bamboo, the primary thickening growth is the growth period from bud to mature shoot before it emerges from the ground, with a significant increase in cross-sectional area during this period [5]. Due to the lack of the interfascicular cambium, the culm diameter and number of internodes of the mature bamboo culm are largely determined by the primary thickening growth of the underground shoot [5,16]. In addition, the primary thickening growth of the underground shoot is important for the biomass and wood properties of bamboo [16]. Therefore, it is of great importance to identify and analyze stable variants of Moso bamboo that have mutations in the primary thickening growth of underground shoots.

*P. edulis* f.* exaurita* is a stable variant of Moso bamboo [17]. Compared to WT Moso, the most striking feature of *P. edulis* f. *exaurita* is its dwarfism. Here, we compared *P. edulis* f.* exaurita* and WT Moso bamboo morphologically, anatomically, and physiologically and showed that *P. edulis* f.* exaurita* is another stable variant with abnormal primary thickening growth, which is valuable for interpreting the molecular mechanisms underlying the primary thickening growth of bamboo.

## 2. Results

### 2.1. Morphological Comparison between Culms of P. edulis f. exaurita and P. edulis

Morphological analysis revealed that *P. edulis* f. *exaurita* is a dwarf variant of *P. edulis*. The part under the first branch of the variant culm was obviously shorter when compared with the part of the WT Moso culm (Figure 1a). The total height (TH) of the variant was significantly lower and was only ~49% of the height of WT (Figure 1c). The part under the first branch of the bamboo culm is the most important part for industrial and handicraft use. Not surprisingly, the height under the first branch (HUFB) of the variant was also significantly lower than that of the WT Moso (Figure 1c). The ratio of HUFB to TH of the variant was also significantly lower than that of the WT Moso (Figure 1d). In addition to the dwarf phenotype, the culm of the variant is also significantly thinner (Figure 1b). The diameter at breast height (DBH) of the variant was significantly smaller than that of WT (~51%) (Figure 1e).

In order to find out the cause of the dwarf phenotype of the variant, the number of internodes in the entire culm of the variant and WT Moso was counted. The length of every internode under the first branch of the variant and of WT Moso was also counted. The mean value of the internode length of the variant was significantly smaller and was 56.8% of the internode length of WT Moso (Figure 2a,b). Internode length under the first branch increased with increasing internode number at WT Moso. In contrast, in the variant, the internode length first increased to the 13th internode and then decreased (Figure 2b). Like internode length, the mean value of internode thickness of the variant was also significantly less than that of WT Moso. The trend of internode thickness with an increase in internode number was similar for the variant and WT Moso. All decreased as the number of internodes increased (from bottom to top) (Figure 2c). In addition, the number of internodes in the culm of the variant was also significantly lower than that of WT (Figure 2d). The number of internodes in the culm of WT Moso ranged from 38 to 52, while the number of internodes in the variant ranged from 26 to 36 (Figure 2d). The number of internodes under the first branch was also counted (Figure 2d). The number of internodes under the first branch of the variant ranged from 7 to 16, which is also much less than that of WT Moso (14~21) (Figure 2d). However, the ratio between the number of internodes under the first branch and the number of internodes in the whole culm was similar for the variant and WT Moso (Figure 2e).

### 2.2. Variations in Culm Structure of P. edulis f. exaurita and P. edulis Culm

To investigate the variation in the culm structure of WT Moso and the variant, six phenotypic traits were used to analyze phenotypic correlation. These phenotypic traits were HUFB, TH, DBH, internode number under the first branch (INUFB), internode length under the first branch (ILUFB), and total internode number (IN) (Table 1). In WT Moso, the correlation between TH and INUFB was significantly positive, suggesting that INUFB increases as TH increases. No significant correlation was found between these two features in the variant. DBH is an important trait for the evaluation of internode structure, and the correlations with HUFB, INUFB, and ILUFB were significant in WT Moso. However, no significant correlation was found between DBH and these traits in the variant (Table 1). Based on these results, it was assumed that culm structure varied in the variant.

### 2.3. Anatomical Comparison between the Internodes of P. edulis f. exaurita and P. edulis

To further compare the internodes of the variant and WT, an anatomical analysis of each internode under the first branch was performed. Three 2-year-old culms of the variant and WT Moso were selected. The lengths and widths of more than 30,000 cells were measured (Figure 3a). The cell length of the internodes in the variant was significantly shorter than that of WT Moso (Figure 3b). The number of cells in the longitudinal direction of the internodes of the WT Moso was significantly larger than that of the variant (Figure 3c). The law that the relative cell number in the longitudinal direction changed with increasing internode number (from bottom to top) was similar for the variant and the WT Moso (Figure 3c). In addition, the cell width of the internodes of the WT Moso was significantly smaller and was 93% that of the variant (Figure 3d). In the radial direction, the number of cells in the internodes of the WT Moso was significantly greater than that of the variant (Figure 3e). However, the regularity with which the relative number of cells in the radial direction changed with increasing internode number was similar in the variant and WT Moso (from bottom to top), both of which follow the linearly increasing model (Figure 3e).

In addition, cross-sections of each internode below the first branching were utilized to compare the vascular bundles of the variant and WT Moso (Figure 4). The vascular bundles in the culm of the variant were symmetrical, making up a classic open vascular bundle similar to that of the WT Moso (Figure 4a). It was found that the size of vascular bundles was similar in the variant and WT Moso (Figure 4b). The regularity with which the size of the vascular bundles changes with increasing internode number (from bottom to top) was also similar in the variant and WT Moso (Figure 4b). When counting and comparing the density of vascular bundles, no significant difference was observed between the variant and WT Moso (Figure 4c). In contrast, the number of vascular bundles was significantly lower in the variant than in WT (Figure 4d).

### 2.4. Correlation Analysis of Cell Structure Characteristics of P. edulis f. exaurita and P. edulis Culm

To investigate the variation in the cell structure of WT Moso and the variant, two phenotypic traits and seven cell structure traits were used for phenotypic correlation analysis (Table 2). These phenotypic traits and cell structure characteristics were internode length (IL), wall thickness (WAT), cell number in the vertical direction (CN (vertical)), cell length (CL), cell width (CW), cell number in the radial direction (CN (radial)), vascular bundle area (VBA), vascular bundle density (VBD), and vascular bundle number (VBN). The results showed that these characteristics were differentially correlated in the variant and WT Moso. In the WT Moso, the correlation between IL and VB as well as CW and CN (radial) was significant, while in the variant, the correlation was not significant. In addition, the correlations between IL and VBN, CN (vertical) and VBN, CN (radial) and VBN, and WAT and VBN were significant in the Moso and variant, but the degrees were different (Table 2).

### 2.5. Auxin and Zeatin Content of the Buds of P. edulis f. exaurita and P. edulis

According to our previous reports, Moso shoots generally pass through six (S1–S6) distinct morphological stages before becoming mature bamboo shoots [5]. S2-stage buds with a plump body are referred to as awakening buds. To investigate whether there is a defect in the primary thickening growth of shoot buds, we measured the levels of auxin and zeatin in the S2-stage buds of *P. edulis* f. *exaurita* and *P*. *edulis* (Figure 5a). Measurements of free 3-indoleacetic acid (IAA) and 3-indolebutyric acid (IBA) were made, but only IAA was detected. Both IAA and zeatin content were significantly lower in the S2 buds of *P. edulis* f. *exaurita* (Figure 5b,c). In *P. edulis* f. *exaurita*, the contents of IAA and zeatin were 2.76 and 1.72 ng/g, which were 83% and 77% of that in the WT Moso bamboo, respectively.

## 3. Discussion

In recent years, the demand for bamboo raw materials has increased rapidly with the development of the bamboo industry. The cultivation and breeding of bamboo plants are becoming increasingly important [5]. However, the flowering characteristic of bamboo is unique, and its growing season is long and unpredictable [2,18]. Traditional cross-breeding is difficult in bamboo breeding [19,20]. With the development of biotechnology, molecular genetic breeding has opened a new avenue for bamboo breeding. Identification of the key genes controlling important traits is crucial for molecular breeding [19,21]. Comparison of the variant and the WT plants at the molecular level will be helpful in identifying key genes [22,23,24].

However, the stable variant of Moso bamboo is rare. *P. edulis* ‘*Pachyloen*’ is a stable variant of Moso bamboo known for its thick culm walls. By comparing the thick-walled variant and the WT Moso, Wei et al. were able to reveal the primary thickening growth pattern of the underground shoots of Moso bamboo and identify some candidate genes involved in the primary thickening growth process [5]. In 2022, Li et al. discovered a network of miRNA, genes, and phytohormones involved in the primary thickening of bamboo shoots by using the thick-walled variant as an experimental material [16]. These research results suggest that the stable variant plays a crucial role in exploring the basic biology and molecular biology of bamboo.

*P. edulis* f. *exaurita* is a stable variant of Moso bamboo. This bamboo species has been little studied. The most intuitive phenotype of this Moso variant is dwarfing [17]. In crop breeding, dwarfing is an important agronomic trait. It was found that dwarfing can increase crop resistance and yield [25,26]. In rice and wheat breeding, the introduction of dwarf variants has greatly increased yields [26]. In bamboo plants, the culm is the main source of bamboo raw material. Culm height is also an important trait for breeding in Moso bamboo. The identification and systematic characterization of the dwarf variant will further our understanding of bamboo culm development. Here, the culm structure of *P. edulis* f. *exaurita* was found to be significantly different from that of WT Moso. The *P. edulis* f. *exaurita* was significantly shorter compared to WT Moso. Based on years of observations, the dwarfism phenotype of *P. edulis* f. *exaurita* is stable. Thus, *P. edulis* f. *exaurita* is a potentially valuable dwarfing material for the study of bamboo culm development.

Moso bamboo is known for its fast growth rate in the unearthed shoot [13]. Little attention has been paid to the primary thickening growth of underground shoots. However, the primary thickening growth of underground shoots is critical throughout the life cycle [5]. In bamboo plants, the secondary thickening growth of the culm is absent due to the lack of interfascicular cambium. The primary thickening growth of underground shoots determines the number and diameter of internodes in new culms [5,16]. Here, the number and diameter of internodes of the dwarf variant were significantly less than those of WT Moso. Thus, the dwarf phenotype of *P. edulis* f. *exaurita* might be due to the abnormal primary thickening growth of the underground shoots. Not only the number of internodes but also the internode lengths of the variant were smaller than those of WT Moso. The cell widths of the variant were larger than those of the WT Moso, but the difference was not significant in some internodes. When the relative cell number was calculated, it was found that the cell numbers in the longitudinal direction of the individual internodes of the variant were significantly smaller than those of the WT Moso. The above results suggest that the shortened internodes of the variant are due to decreasing cell number. Similar results were found in research on the internode development of a slow-growing bamboo variant, *P. japonica* var. *tsutsumiana* [10]. In addition, the culm wall of the variant internodes was found to be significantly thinner than that of WT Moso. The cell number in the radial direction of the variant decreased compared with the WT Moso. The number of vascular bundles was also significantly lower than in the WT Moso. In *P. edulis* ‘*Pachyloen*’, the abnormal primary thickening growth of underground shoots was found to be responsible for the thickened culm wall [5]. The factor responsible for the dwarfing of the plants is diverse. Based on the above results, the dwarfing of *P. edulis* f. *exaurita* could be caused by the inhibition of the primary thickening growth of underground shoots.

Phytohormones, such as IAA, gibberellin (GA), and abscisic acid (ABA), are important regulators of plant growth and development. Phytohormones also play an important role during the primary thickening growth of Moso bamboo shoots. For example, in the thick-walled variant (*P*. *edulis ‘Pachyloen’*) with a larger SAM and abnormal primary thickening growth, a comparison of transcriptome data from buds in the S2 stage revealed that a large number of genes involved in auxin transport and signal transduction are upregulated [5]. Moreover, a defect in the auxin signaling pathway can lead to abnormality in primary thickening growth and eventually to a dwarfing phenotype in bamboo, such as in *P*. *nidularia* f. *farcta*, a dwarf variant of *P*. *nidularia* with a smaller SAM, since a lower auxin concentration was found in its buds [9]. In addition, zeatin was important in modulating plant architecture [27]. Higher zeatin content was also found in *P. edulis* ‘*Pachyloen*’, a bamboo variant with abnormal primary thickening growth [16]. Here, significantly lower levels of IAA and zeatin were found in the buds of *P*. *edulis* f. *exaurita*, suggesting that the primary thickening growth of *P*. *edulis* f. *exaurita* is also abnormal. The lower levels of auxin and cytokinin could lead to the abnormal development of internodes and nodules during the primary thickening growth of *P*. *edulis* f. *exaurita*, eventually causing the dwarfed culms and internodes with thinner internode walls of *P*. *edulis* f. *exaurita*.

In conclusion, *P. edulis* f. *exaurita* is a stable dwarf variant of Moso bamboo. Through morphological and anatomical investigation on the culm and measurement of hormones in buds in S2, the *P. edulis* f.* exaurita* was thought to be a variant with inhibited primary thickening growth. This work will provide valuable information for further exploration of the molecular mechanisms underlying the primary thickening growth of underground shoots and molecular breeding of Moso bamboo.

## 4. Materials and Methods

### 4.1. Morphological Analysis of the Variant Culm and the WT Moso

Samples of Moso variant and WT Moso bamboo were collected from the bamboo garden of Special Bamboo Species Breeding Farm in Changzhou city, Jiangsu Province, China (120°11′ N,31°78′ E). More than 20 culms of the variant and WT Moso bamboo were collected to measure total height, height under the first branch, number of internodes, internode length, and diameter at breast height.

### 4.2. Cellular Analysis of Variant Internodes and WT Moso

Three 2-year-old mature culms of the variant and WT Moso were selected as experimental materials, respectively. The 1 cm wide sections in the center of the internodes were collected and fixed in formalin–acetic acid-70% alcohol (FAA, *v*/*v*) to make slices. The 1 cm wide sections of every internode under the first branch of the variant and WT Moso were collected. More than 105 sections were collected and fixed. Then, these 1 cm sections were fixed for at least 48 h. The processes of dehydration, infiltration, paraffinization, and staining were performed according to a previous method [10]. The section pictures were taken by using a Leica DM2500 light microscope (Leica, Wetzlar, Germany). Then, ImageJ software was used to measure the length and width of the cells. Cell length refers to the straight-line distance from one end to the other end in the longitudinal orientation. Cell width refers to the straight-line distance from one end to the other end at the transverse orientation. Here, a total of 108 sections from the internodes under the first branch of the variant and Moso bamboo were used for images. More than 300 cells in each image were observed and counted.

### 4.3. Analysis of the Vascular Bundles of the Internode of the Variant and WT Moso

Cross-sections of the culm walls were prepared using the method described previously [10]. Photoshop was used to stitch together photos of different parts of the culm wall. The number and area of vascular bundles were counted and measured using a Leica DM2500 light microscope (Leica, Wetzlar, Germany) and ImageJ software. The parts marked with black lines in Figure 4a were used to measure the area of vascular bundles. A total of 78 sections from the internodes under the first branch of the variant and Moso bamboo were used for the images. More than 10 vascular bundles were counted and measured on each image.

### 4.4. Quantification of Hormones

To quantify the content of auxin (IAA and IBA) and zeatin in S2-stage buds, three biological replicates were used for measurement. Because S2 buds are very small, each replicate contains more than three buds. Endogenous IAA, IBA, and zeatin were extracted, purified, and detected by methods described previously [28].

### 4.5. Statistical Analysis

Analysis of differences in culm height, internode number, internode length, etc., between the variant and WT Moso was performed using integrated *t*-test analysis with GraphPad Prism version 8.0 (GraphPad Software, San Diego, CA, USA). All linear or nonlinear regression analyses were performed with GraphPad Prism version 8.0 using the integrated formula with default argument.

## Figures and Tables

**Figure 1 plants-12-01759-f001:**
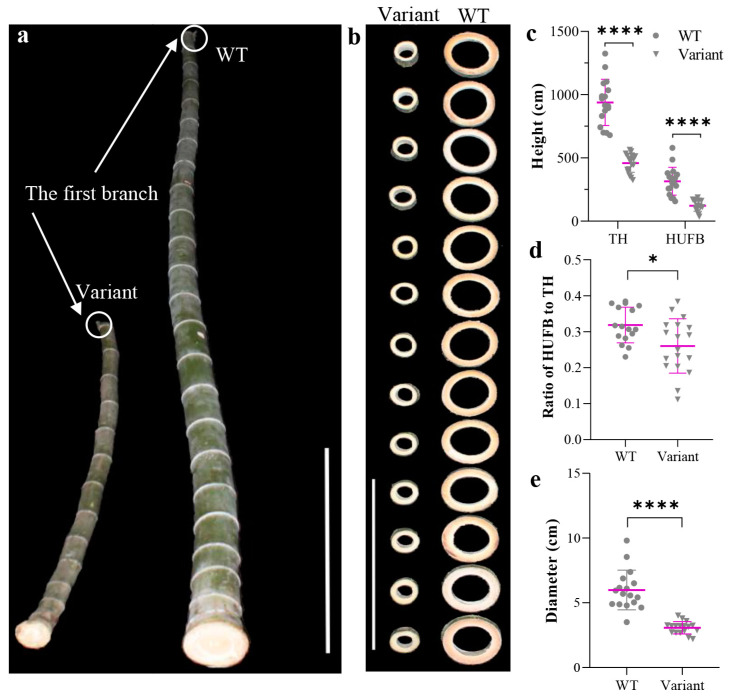
Morphology of *P. edulis* f. *exaurita* (Variant) and *P. edulis* (WT Moso). (**a**) Morphology of 2-year-old culms (parts under the first branch of the culm) of WT Moso and variant plants. The positions of the first branch at the culm are marked with white circles. The white line represents 100 cm. (**b**) Radial morphology of internodes under the first branch of WT Moso and variant plants. The white line represents 30 cm. (**c**) Total height (TH) and height under the first branch (HUFB) of WT Moso and variant plants. (**d**) Ratio of HUFB to TH of WT Moso and variant plants. (**e**) Diameter at breast height of WT Moso and variant plants. * *p* < 0.05, **** *p* < 0.0001.

**Figure 2 plants-12-01759-f002:**
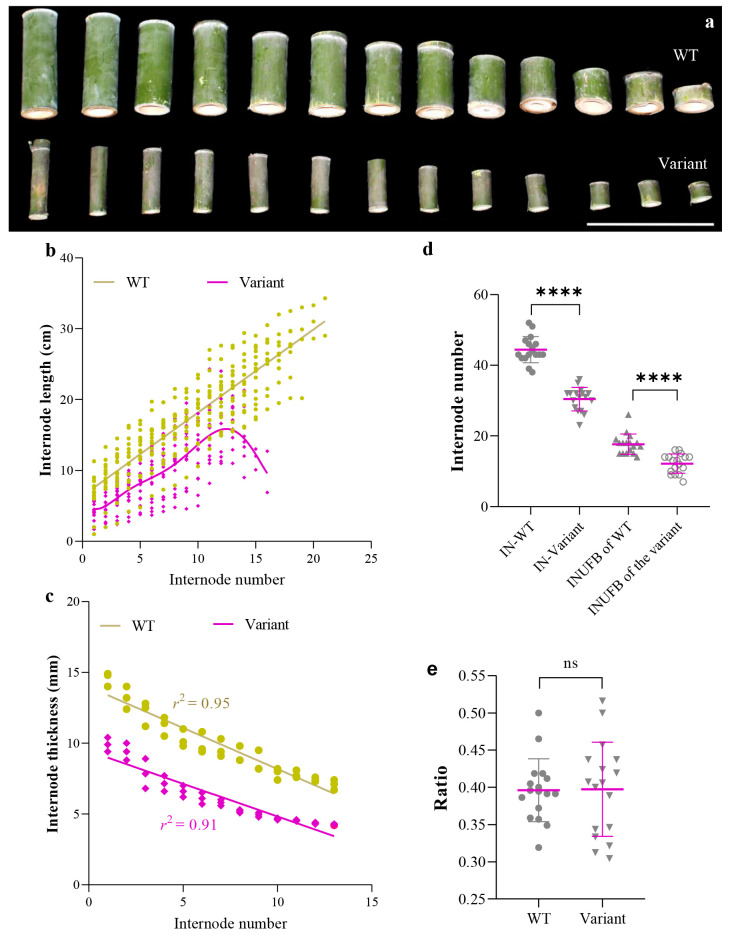
Culm internode of *P. edulis* f. *exaurita* (variant) and *P. edulis* (WT Moso). (**a**) Morphology of internodes under the first branch of WT and variant plants. (**b**) Internode length of WT Moso and the variant. (**c**) Internode thickness of WT Moso and the variant. (**d**) Number of internodes of the whole culm and internodes under the first branch of WT Moso and the variant. (**e**) Ratio of internode number under the first branching (INUFB) to total internode number (IN) of WT Moso and the variant. The white line in (**a**) represents 30 cm. **** *p* < 0.0001. ns, no significant.

**Figure 3 plants-12-01759-f003:**
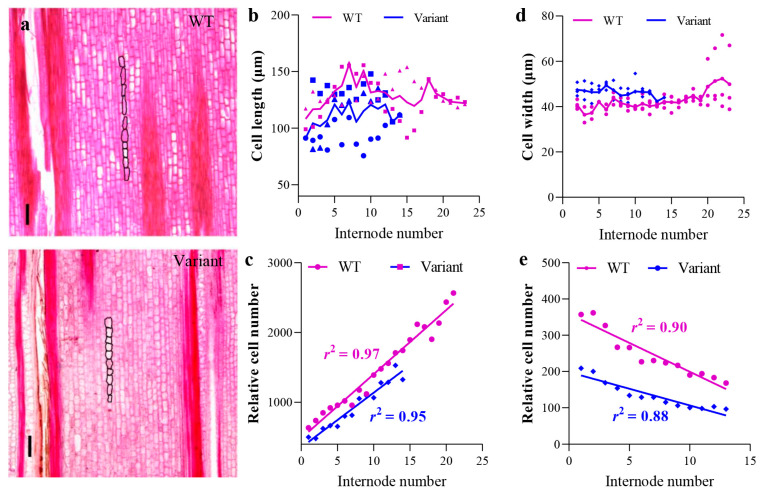
Cellular characterization of *P. edulis* f. *exaurita* (variant) and *P. edulis* (WT Moso). (**a**) Cells of the third and eleventh mature internodes of 2-year-old culms of WT Moso and the variant. (**b**) Length of internode parenchyma cells of WT Moso and the variant. (**c**) Cell number in longitudinal direction of internodes of WT Moso and the variant. (**d**) Width of internode parenchyma cells of WT Moso and the variant. (**e**) Cell number in radial direction of internodes of WT Moso and the variant. The black line in (**a**) represents 200 μm.

**Figure 4 plants-12-01759-f004:**
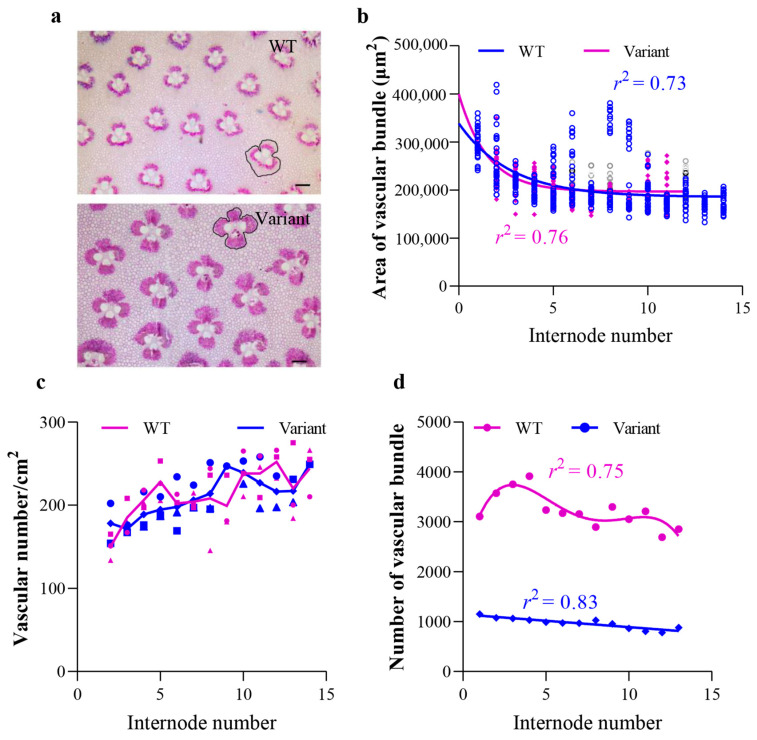
Vascular bundles of *P. edulis* f. *exaurita* (variant) and *P. edulis* (WT Moso). (**a**) Cross-sections of the culm walls of WT Moso and the variant plants. The enclosed black lines represent the outlines of the vascular bundles. (**b**) Aera of the vascular bundles of WT Moso and the variant. (**c**) Density of vascular bundles of WT Moso and the variant. (**d**) Number of vascular bundles of WT Moso and the variant. Black line in (**a**) represents 200 μm.

**Figure 5 plants-12-01759-f005:**
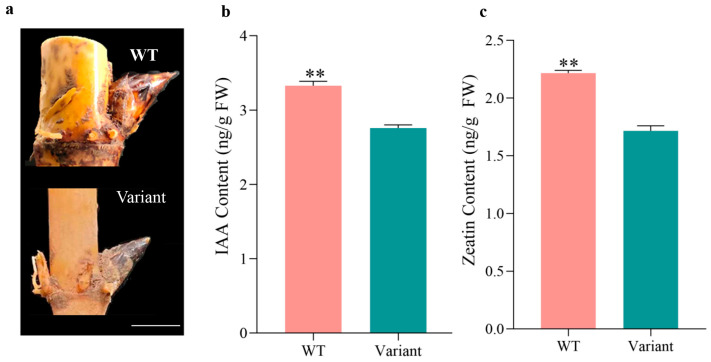
Auxin and zeatin content of S2-stage buds of *P. edulis* f. *exaurita* (variant) and *P. edulis* (WT Moso). (**a**) S2-stage buds of WT Moso and the variant plants. (**b**) IAA content of S2 buds of WT Moso and the variant. (**c**) Zeatin content of S2 buds of WT Moso and the variant. ** *p* < 0.01. White line in (**a**) represents 1 cm.

**Table 1 plants-12-01759-t001:** Correlation analysis of 6 phenotypic traits in *P. edulis* f. *exaurita* (variant) and *P. edulis* (WT Moso).

Variant						
	HUFB	TH	DBH	IN	INUFB	ILUFB
HUFB	1 ****	0.67 **	0.36	0.61 **	0.68 ****	0.71 ***
TH		1 ****	0.70 **	0.67 **	0.40	0.51 *
DBH			1 ****	0.60 **	0.28	0.163
IN				1 ****	0.77 ****	0.12
INUFB					1 ****	0.28
ILUFB						1 ****
WT						
	HUFB	TH	DBH	IN	INUFB	ILUFB
HUFB	1 ****	0.92 ****	0.87 ****	0.70 **	0.90 ****	0.89 ****
TH		1 ****	0.85 ****	0.68 **	0.80 ****	0.86 ****
DBH			1 ****	0.82 ****	0.78 ****	0.76 ****
IN				1 ****	0.77 ****	0.48
INUFB					1 ****	0.61 *
ILUFB						1 ****

Notes: HUFB: height under the first branch, TH: total height, DBH: diameter at breast height, IN: internode number, INUFB: internode number under the first branch, ILUFB: internode length under the first branch. **** represents 0.01% prominence level, *** represents 0.1% prominence level, ** represents 1% prominence level, * represents 5% prominence level.

**Table 2 plants-12-01759-t002:** Correlation analysis of 2 phenotypic traits and 7 cell structure traits in *P. edulis* f. *exaurita* (variant) and *P. edulis* (WT Moso).

WT									
	IL	CN (Vertical)	CL	CW	CN (Radial)	WAT	VBA	VBD	VBN
IL	1 ****	0.99 ****	0.51	0.43	−0.96 ****	−0.99 ****	−0.81 ***	0.80 ***	−0.64 *
CN (vertical)		1 ****	0.45	0.41	−0.95 ****	−0.98 ****	−0.80 ***	0.79 ***	−0.65 *
CL			1 ****	0.35	−0.62 *	−0.60 *	−0.49	0.45	−0.27
CW				1 ****	−0.60 *	−0.43	−0.32	0.26	−0.40
CN (radial)					1 ****	0.98 ****	0.81 ***	−0.79 ***	0.60 *
WAT						1 ****	0.84 ****	−0.83 ****	0.58 *
VBA							1 ****	−0.89 ****	0.20
VBD								1 ****	−0.075
VBN									1 ****
Variant									
	IL	CN (vertical)	CL	CW	CN (radial)	WT	VBA	VBD	VBN
IL	1 ****	0.98 ****	0.62 *	−0.54	−0.92 ****	−0.94 ****	−0.58	0.74 **	−0.91 ****
CN (vertical)		1 ****	0.54	−0.53	−0.94 ****	−0.96 ****	−0.64 *	0.83 **	−0.84 ***
CL			1 ****	−0.14	−0.67 *	−0.66 *	−0.63	0.35	−0.72 *
CW				1 ****	0.32	0.43	0.05	−0.34	0.46
CN (radial)					1 ****	0.99 ****	0.79 **	−0.85 ***	0.80 **
WT						1 ****	0.76 **	−0.85 ***	0.81 **
VBA							1 ****	−0.74 **	0.42
VBD								1 ****	−0.43
VBN									1 ****

Notes: IL: internode length, CN (vertical): cell number in the vertical, CL: cell length, CW: cell width, CN (radial): cell number in the radial, WAT: wall thickness, VBA: vascular bundle area, VBD: vascular bundle density, VBN: vascular bundle number. **** represents 0.01% prominence level, *** represents 0.1% prominence level, ** represents 1% prominence level, * represents 5% prominence level.

## Data Availability

In this section, transcriptional data were downloaded from NCBI, and physiological and anatomic metabolic data were measured by the authors themselves.

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
