# Peer review of "Morphological and Anatomical Analysis of the Internodes of a New Dwarf Variant of Moso Bamboo, Phyllostachys edulis f. exaurita"

_plants, 2023, doi:10.3390/plants12091759_

Round 1

Reviewer 1 Report (New Reviewer)

Stable mutants is important for basic biology research in bamboo. This manuscript ‘Morphological and anatomical analysis of the internodes of a new dwarf variant of Moso bamboo, Phyllostachys edulis f. exaurita’ systematically compared the stable variant and the WT Moso bamboo morphologically and anatomically. Contents of IAA and zeatin in the buds were also detected.

These results provide valuable information for Moso bamboo research. The subject is interesting. The experiments were well described, designed and conducted. Data was well analyzed and presented. I have some comments and would recommend the work for publication in Plants after minor revise.

1. Lines 24-25: ‘Furthermore, lower IAA and zeatin concentrations were detected in the buds of the variant.’ The font format should be unified

2.Lines 85-86:’The part under the first branch (HUFB) of the bamboo culm is the most important part for industrial and handicraft use.’ HUFB should be the abbreviation of “height under the first branch”.

3. Line241: “Culm height is also an important trait for breeding Moso bamboo.” Some grammar errors. “for breeding in Moso bamboo”

4. Lines 246-247: “Thus, P. edulis f. exaurita is a potentially valuable dwarfing material for on the study of bamboo culm development.” Some grammar errors.

5. Lines 259-261: “When the relative cell number was calculated, it was found that the cell numbers of the individual internodes of the variant were significantly smaller than those of the WT Moso.” What cell does the cell number refer to? In the longitudinal direction of the internodes or the radial direction? Or both in the longitudinal and radial direction?

6. At “4.2. Cellular analysis of variant internodes and WT Moso”. Culms used for experiment were 1-year-old. However, in figure lends of Fig 1 and Fig 3, the culms were 2-years-old. Please make sure which age culms were used here.

7. Keywords

Add a key word “primary thickening growth”.

Author Response

Reviewer 2 Report (New Reviewer)

In my opinion, the article entitled: “Morphological and anatomical analysis of the internodes of a new dwarf variant of Moso bamboo, Phyllostachys edulis f. exaurita”, despite the applied corrections, is not suitable for publication in its current version. The introductory chapter definitely needs to be improved, it is unprofessional and chaotic. A chapter Conclusions summarizing the greatest achievements of the authors should be added to the article.

Specific comments:

Title: The abbreviation "f." should not be in italics!

The first paragraph of the Introduction chapter should be thoroughly revised, it is not coherent.

Line 31: Modify as:  Bamboos belong to the subfamily Bambusoideae of the grass family Poaceae.

Line 31-32: Please expand and explain the sentence, citing references is not enough: “Compared to other members of Poaceae, most bamboos are woody and taller.”

Line 32-33: “Its excellent ability to sequester carbon contributes greatly to climate change mitigation.” Please provide references!

Line3 36-37: “The estimated value will reach about $100 billion in 2025 (www.grandviewresearch.com/industryanalysis/bamboos-market).” - Is this sentence needed?

Line 93: Please describe exactly what Fig. 1a shows, what do the authors think can be seen in this photograph?

Lines 216-226: Please modify the first paragraph of the Discussion and remove information that is redundant in the context of the article.

The text is understandable, of course it can still be improved linguistically.

Author Response

Reviewer 3 Report (New Reviewer)

The article "Morphological and anatomical analysis of the internodes of a new dwarf variant of Moso bamboo, Phyllostachys edulisf. exaurita" by Ruofei Zha, Tianguo Chen, Qingnan Liu, Qiang Wei, Feng Que considers the possibility of using a new dwarf variant of bamboo in breeding and technology. In general, the work is written clearly and contains all the necessary parts, with the exception of the Conclusion section. However, some things are obscure. Perhaps the authors meant something else (?): long flowering period of bamboo plants is one of the limiting factors inhibiting research progress in culm development of bamboo plants. The problems of breeding and creating hybrids or mutants are associated with long periods between flowering and not long flowering.

It also requires clarification of what the authors mean by culm and internodes under the first branch of WT Moso and the variant, since this is not obvious without a photo of the plants.

The same applies to Figure 3, since it is not clear how many measurements were made, how many stems the authors used. It is also not obvious that the identified trends are confirmed in other segments. This requires a more thorough description in the materials and methods section and in the results section.

It is rather strange that there is no information about the features of branching, the arrangement of leaves, and with a rather large section devoted to roots, it is not clear why there is no photo and morphological analysis of root systems.

It is also not clear why the authors did not draw up a conclusion and did not explain how exactly this study can be used and what is its fundamental novelty.

If these inaccuracies are corrected, the work may be accepted.

Round 2

Reviewer 3 Report (New Reviewer)

Manuscript Morphological and anatomical analysis of the internodes of a new dwarf variant of Moso bamboo, Phyllostachys edulis f. exaurita by Ruofei Zha et al. has been substantially revised and is available for publication in its present form. The work is interesting and has great prospects for development.

This manuscript is a resubmission of an earlier submission. The following is a list of the peer review reports and author responses from that submission.

Round 1

Reviewer 1 Report

23/02/23

Dear Editor,

I reviewed the manuscript entitled Morphological and anatomical analysis of the internodes of a new dwarf variant of 2 Moso bamboo, Phyllostachys edulis f. exaurita, by Zha et al.

The manuscript tries to investigate the dwarf mutant of Moso bamboo from morphological and anatomical point of view, focusing on internodes characteristics. The work presents some weaknesses, especially in the results explanation and data analysis.

What do the author specifically mean with “thickening growth”? this must be specified.

Line 56: the largest area -> the largest cultivation area

Line 57-70 need to be rephrased

In Material and methods, from each of the three plants, how many sections were used? and how many pictures were analyzed?Only one? Where did this 30000 cells come from?

Line 274: is “9”  a reference?

Results:

Par 2.1 I think the figure citations are wrong, fig 1b and 1d do not refer to the right figure.

line99: the law of change?? What do you mean?

Fig.2 legend: rule of… what do you mean?

Par 2.2 Showing only correlation analysis, the reader can make no idea of the value of each of the six parameters. Are you sure lines 128-131 are correct? Try to explain it better please.

How many vascular bundles were counted and measured?

Figure 4 b and d: vascular and not vacular.

Figure 4 a is in very low quality, can you perform a nicer sectioning and image acquisition?

Reviewer 2 Report

Bamboo is an important forestry resource in the world and has great economic and ecological value. The lack of mutants due to the long flowering period of bamboo plants is one of the limiting factors inhibiting research progress in culm development of bamboo plants. A systematic comparison between the stable variant and the WT bamboo plant is helpful to promote basic research on bamboo biology. Zha et al. investigated and compared the morphological and anatomical traits of a dwarf variant of Moso bamboo, Phyllostachys edulis f. exaurita with   Moso bamboo. These results definitely provide valuable information for further exploration. However, I still have the following concerns that should be considered by the authors.

Major concerns

1.  In morphology, please add the ratio of the height under the first branch to plant height.

2.  In anatomy, the authors only compared the cells in the elongated region and did not consider the cell characteristics of the cell division region, which is very important for bamboo internode development.

3. I suggest that if there is a sufficient number of mutants, it is best to determine the ordinal number of stable internode of the variety for comparison with Moso bamboo.

Minor concerns

1. Please note the correctness of the format of the references in P10, lines 274 and 281. 

2. Please note the completeness of the reference information in P11, Ref. 17

Round 2

Reviewer 2 Report

The author has answered my concerns. Expect the results of the work on the cell division region of the mutant to be presented as soon as possible。